# Structure-based discovery of small molecules that disaggregate Alzheimer's disease tissue derived tau fibrils in vitro

Paul M. Seidler[1,2,3,4,5,6,10], Kevin A. Murray[1,2,3,4,5,10], David R. Boyer[1,2,3,4,5,10], Peng Ge[1,2,3,4,5], Michael R. Sawaya[1,2,3,4,5], Carolyn J. Hu[1,2,3,4,5], Xinyi Cheng[1,2,3,4,5], Romany Abskharon[1,2,3,4,5], Hope Pan[1,2,3,4,5], Michael A. DeTure[7], Christopher K. Williams[8], Dennis W. Dickson[7], Harry V. Vinters[8,9] & David S. Eisenberg[1,2,3,4,5] ✉

Alzheimer's disease (AD) is the consequence of neuronal death and brain atrophy associated with the aggregation of protein tau into fibrils. Thus disaggregation of tau fibrils could be a therapeutic approach to AD. The small molecule EGCG, abundant in green tea, has long been known to disaggregate tau and other amyloid fibrils, but EGCG has poor drug-like properties, failing to fully penetrate the brain. Here we have cryogenically trapped an intermediate of brain-extracted tau fibrils on the kinetic pathway to EGCG-induced disaggregation and have determined its cryoEM structure. The structure reveals that EGCG molecules stack in polar clefts between the paired helical protofilaments that pathologically define AD. Treating the EGCG binding position as a pharmacophore, we computationally screened thousands of drug-like compounds for compatibility for the pharmacophore, discovering several that experimentally disaggregate brain-derived tau fibrils in vitro. This work suggests the potential of structure-based, small-molecule drug discovery for amyloid diseases.

Recognizing that Alzheimer's disease (AD) is the most common neurodegenerative disease worldwide, increasingly prevalent in our aging population, scientists have sought therapeutic interventions capable of slowing disease progression. Although cognitive decline and brain atrophy correlate with tau fibril formation in AD[1–3], finding effective therapeutics that enter the brain, enter neurons that house tau fibrils and break down the fibrils remains a challenge. Considerable efforts have been made to diminish tau aggregation in AD, including stabilizing microtubules, inhibiting tau phosphorylation or acetylation, reducing tau expression, or inhibiting fibrilization[4]. Tau aggregation

inhibitors including curcumin and methylene blue/LMTX have progressed to Phase II and Phase III clinical trials, respectively, but have yet to show efficacy in treating disease[5]. Here we pursue a new approach. We introduce structure-based screening to identify small molecules capable of disaggregating tau amyloid fibrils as the first step on a new therapeutic route for Alzheimer's disease.

Despite the unusual stability of pathogenic amyloid fibrils, most robustly stable in SDS, urea, guanidinium, and at elevated temperatures, a few small molecules are known to disaggregate fibrils in aqueous conditions into smaller non-toxic species. Prominent among

[1]Department of Chemistry and Biochemistry, UCLA, Los Angeles, CA, USA. [2]Department of Biological Chemistry, UCLA, Los Angeles, CA, USA. [3]UCLA-DOE Institute, Los Angeles, CA, USA. [4]Molecular Biology Institute, UCLA, Los Angeles, CA, USA. [5]Howard Hughes Medical Institute, Los Angeles, CA, USA. [6]Department of Pharmacology and Pharmaceutical Sciences, University of Southern California, Los Angeles, CA, USA. [7]Department of Neuroscience, Mayo Clinic, Jacksonville, FL, USA. [8]Department of Pathology and Laboratory Medicine, David Geffen School of Medicine, UCLA, Los Angeles, CA, USA. [9]Department of Neurology, David Geffen School of Medicine, UCLA, Los Angeles, CA, USA. [10]These authors contributed equally: Paul M. Seidler, Kevin A. Murray, David R. Boyer. ✉e-mail: david@mbi.ucla.edu

these is epigallocatechin gallate (EGCG), a polyphenolic compound found in green tea (Fig. 1a) and extensively investigated with similar compounds in some 4000 research papers[6–14]. However, EGCG has not proven an effective drug, perhaps because of limited bioavailability (particularly in the brain), promiscuous protein binding, and ready modification in bodily fluids[15]. Unlike other small molecules that inhibit tau fibril formation, EGCG disaggregates previously formed fibrils, indicating that visualization of EGCG bound to the fibrils may be possible before disaggregation.

To illuminate the remarkable but so far unexplained mechanism of EGCG's disaggregation of tau amyloid, we have cryogenically trapped an intermediate on the pathway of EGCG-driven disaggregation of tau fibrils extracted from post-mortem brains of AD patients. The EGCG-AD-tau-fibril complex reveals what we term the EGCG pharmacophore on tau fibrils. We use this pharmacophore for in silico screening of a library of drug-like small molecule compounds with properties predictive of central nervous system penetration. We find by experiment that several of our predicted tau binders disaggregate AD-tau, suggesting the power of our structural approach to compound discovery. Tau pathology likely propagates throughout the brain by prion-like seeding, in which aggregates in one diseased cell travel to adjacent cells and induce further protein aggregation[16–18]. Thus, we also characterize the cytotoxicity and prion-like seeding capacity of AD-tau fibrils after disaggregation by our lead small molecule disaggregants. Our analysis of the EGCG–AD-tau–fibril complex suggests a plausible means by which a small molecule can disassemble stable fibril architecture, informing future studies of potential therapeutics.

## Results

### Structure determination of tau fibrils in complex with EGCG

We surveyed the time course of EGCG-driven tau disaggregation by incubating AD brain-derived tau fibrils with EGCG for 1, 3, 6, and 24 h at 37 °C. We monitored the reduction of AD tau aggregates over time by dot blot analysis using the monoclonal antibody GT38, which specifically recognizes AD tau fibrils (Supplementary Fig. 1a)[19]. We observed a reduction in aggregate level beginning at 1–3 h, while total hyperphosphorylated tau levels remained the same, as determined by dot blot with antibody AT8 (Supplementary Fig. 1b). In agreement with these observations, negative stain electron micrographs of the 3-h fibrils appear largely intact but somewhat swollen, and at later times the fibrils disappear and are nearly gone after 24 h of incubation (Fig. 1b). Thus, intermediates in fibril disassembly appear to be most abundant at time points up to 3 h, according to two lines of evidence.

To resolve the tau-EGCG interactions responsible for tau fibril disassembly, we collected and processed cryoEM images at the 1- and 3-h time points identified above as being enriched in disassembly intermediates (Supplementary Fig. 2). As a negative control, we also collected and processed images of AD tau fibrils before incubation with EGCG (Supplementary Fig. 2). Helical reconstructions of all three datasets revealed the Paired Helical Filament (PHF) tau polymorph, while Straight Filaments, a polymorph of tau fibrils usually found in AD at low abundance, were not present in sufficient quantities for structure determination (Fig. 1c, d and Supplementary Fig. 3). Supplementary Fig. 3d also shows a difference map between the 0- and 3-h maps.

Three new densities alongside the PHF (Fig. 1d; Sites 1-3) were revealed in cryoEM maps of PHFs incubated with EGCG for 1- and 3-h. These new densities are not present in the map of our control, lacking EGCG (Fig. 1c). Reassuringly, our control map precisely matches previously published tau PHF maps (Fig. 1c)[20,21]. Moreover, the density attributed to tau exhibits the same backbone path at all time points (Fig. 1c, d and Supplementary Fig. 3a–c). Consequently, conformational changes among our refined atomic models of the PHFs are minor (Supplementary Fig. 4a, b), and our discussion of EGCG's interaction with tau fibrils focuses on Sites 1-3 in the 3-h structure, where EGCG density appears most strongly.

The density at Site 1 is the most prominent of the three sites. It lies in a cleft formed by the junction of two tau protofilaments bordered by the polar residues Asn327, His329, Glu338, and Lys340. It has three distinct lobes, which resemble the three aromatic branches of EGCG, leading us to attribute it to EGCG (Fig. 1d, f). Sites 2 and 3 are located alongside the β-helix region of the fibril core, adjacent to Lys321 and Lys317, respectively. Densities at Sites 2 and 3 are smaller than Site 1 and do not have the characteristic three-lobe shape of EGCG. Moreover, an atomic model of EGCG bound to Site 1 buries a much greater area on the PHF surface than does EGCG modeled at Sites 2 and 3 ($227\,\text{Å}^2$ vs. $34\,\text{Å}^2$ and $38\,\text{Å}^2$, respectively) (Supplementary Fig. 5). Therefore, we focus on Site 1 as the main EGCG pharmacophore.

Our atomic model shows EGCG molecules stacking in two helical columns that span the fibril length: one column at each of the two symmetry-related inter-protofilament clefts. A view perpendicular to the PHF fibril axis shows that Site 1 density repeats every 4.8 Å, in register with the spacing between tau molecules (Fig. 1e–g). To model EGCG molecules in such closely spaced densities, we adjusted the EGCG conformation to be nearly planar, thus avoiding a steric clash with neighboring EGCG molecules in the stack. EGCG stacking has been observed previously in EGCG crystal form IV[22]. However, the stacking distance in the crystal structure is nearly 1 Å greater than the 4.8 Å amyloid spacing—a difference accommodated by a greater tilting of the EGCG molecular plane away from the stacking axis' normal plane. It is clear that EGCG complexed with tau does not stack at the crystallographic distance. If that were the case, enforcement of helical symmetry during the cryoEM refinement would have produced continuous density across layers. The clean 4.8 Å separation of Site 1 densities suggests that EGCG molecules form a 1:1 complex with tau molecules in the fibril.

To establish the most likely binding pose of EGCG in the 3-h map, we assessed 6 different conformations (A–F) using numerous metrics, including the fit of the model to the density, the buried surface area, and the number of hydrogen bonds formed. Conformation C (Supplementary Fig. 6 and Supplementary Table 2) scored moderately better than the others and features multiple stabilizing interactions with the protein side chains. In this pose, the 4′ hydroxyl of the EGCG D-ring hydrogen bonds with His329 and Glu338, and the 3′ and 4′ hydroxyl from the monocyclic B-ring hydrogen bonds with Asn327 (Fig. 1f). A fourth hydrogen bond is made between Lys340 and a hydroxyl from the A ring moiety of EGCG. Partial π–π stacking adheres EGCG ring D to the aromatic side chain of His329 (Supplementary Fig. 6c).

In summary, EGCG disaggregates AD-tau PHFs over the course of 24 h, and by structure determination of fibrils after 3 h of EGCG incubation, we trap an intermediate on the disassembly pathway. This structure reveals EGCG molecules stacked in the two symmetrically related clefts formed at the junction of the two protofilaments. We consider the EGCG binding site on AD-tau fibrils as an EGCG pharmacophore.

### Screening for tau disaggregants using the EGCG pharmacophore

EGCG itself is a poor therapeutic candidate owing to its polyphenolic molecular structure, which results in unfavorable drug-like properties and restricts brain penetration. Fortunately, our structure provides clues for discovering new disaggregant molecules with more desirable drug characteristics. We hypothesized that effective disaggregants can be identified by computationally selecting molecules complementary in shape to the pharmacophore defined by the trapped EGCG-AD-tau.

For in silico docking, we selected two small molecule libraries: (1) currently FDA-approved small molecule drugs (~1700 compounds), and (2) ChemBridge CNS-set, a ~60,000 compound library containing drug-like compounds that have characteristics favoring brain-penetration and oral bioavailability (following the Lipinski rule of

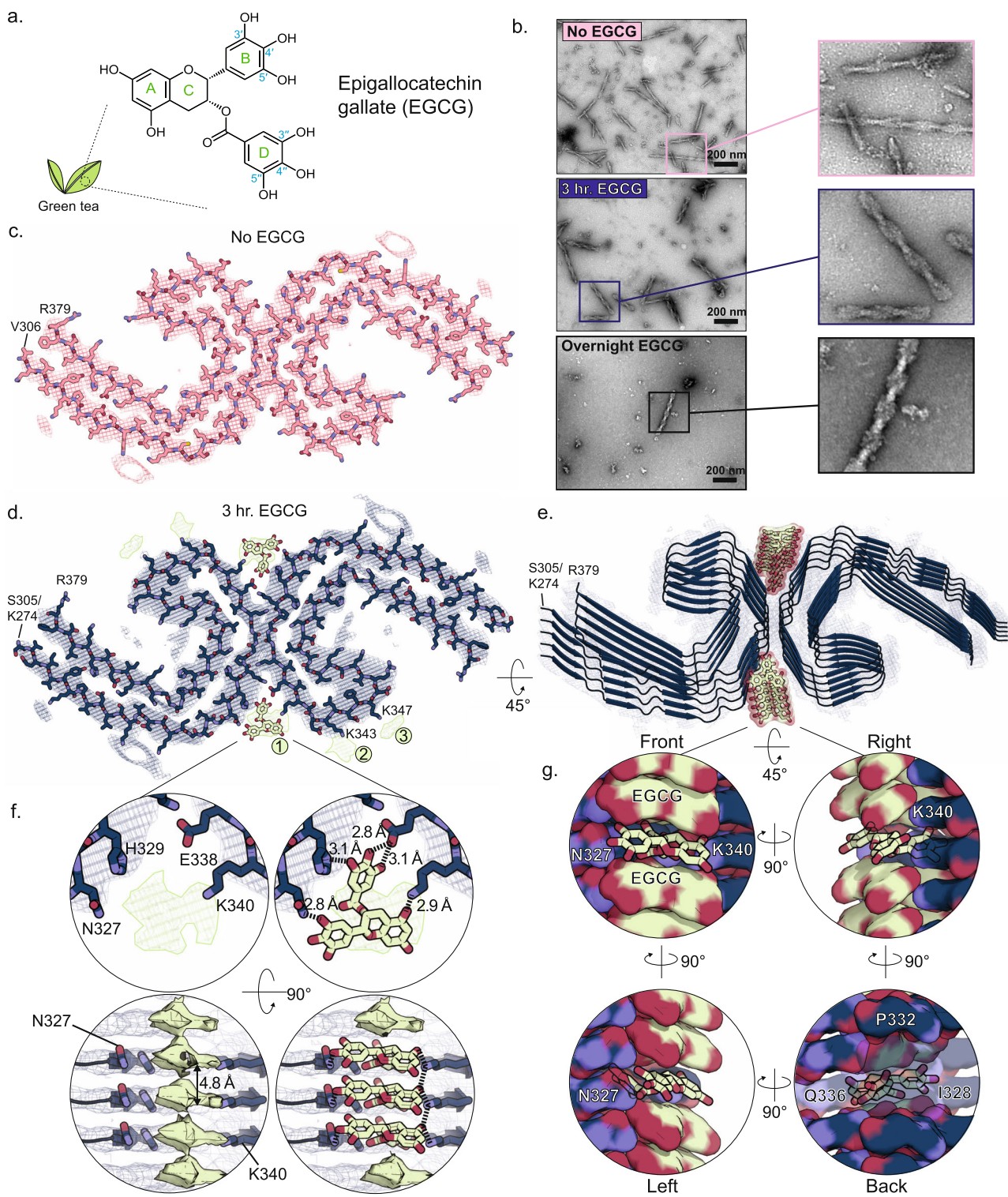

**Fig. 1 | CryoEM structure of AD-tau PHF in complex with EGCG.**
**a** Epigallocatechin gallate (EGCG) is the most abundant polyphenol in green tea. It includes a benzenediol ring (A) adjoined to a tetrahydropyran moiety (C), which are connected to a galloyl ring (D) and pyrogallol ring (B). The 8 hydroxyl groups allow EGCG to engage in hydrogen bonding and other polar interactions with numerous biomolecules. **b** Electron micrographs of brain-derived PHFs over the course of EGCG incubation. Without EGCG (top), numerous fibrils are observed (representative image from $n = 15$). After 3-h incubation at 37 °C (middle), subtle changes in the fibril morphology are present, with a widening of the fibril fuzzy coat (representative image from $n = 15$). Far fewer fibrils are seen at this time point. After overnight EGCG incubation (bottom), the rare remaining fibrils appear swollen and disturbed (representative image from $n = 8$). **c** Cross-sectional view of the AD patient brain-derived tau PHF cryoEM structure before the addition of EGCG. **d** Tau PHF structure

following 3-h incubation with EGCG. Three new regions of density become apparent with the addition of EGCG (Sites 1–3). Site 1 is located in the polar cleft at the intersection of the two protofilaments composing the PHF. Sites 2 and 3 of new density are observed adjacent to K343 and K347 near the β-helix of the fibril. Both Sites 2 and 3 display weaker density than Site 1. **e** Tilted view of the 3-h structure with EGCG bound at Site 1. **f** Close up top- and side-views of EGCG in Site 1. This region borders N327, H329, E338, and K340 of the fibril, with EGCG making polar and hydrogen-bond contacts with these residues. EGCG adopts a primarily planar conformation when bound to the fibril, stabilized by π-π interactions of the stacked aromatic rings of EGCG. When viewed from the side of the fibril axis, the EGCG density is seen stacking with the same period as the fibril layers. **g** Side-view of a single EGCG molecule buried by the fibril and other copies of EGCG.

five, low polar surface area, etc.). EGCG was included as a positive control in both in silico docking libraries. We used two docking methods; AutoDock Vina[23] and RosettaLigand[24] to identify compounds that could bind favorably in EGCG binding Site 1 (Fig. 2a). Both docking methods ranked EGCG among the top scoring compounds, with predicted binding energies >2 standard deviations stronger than the average compound (Fig. 2c, b). This result indicates that both docking methods successfully recognize EGCG as a ligand of AD tau fibrils.

From the in silico screen, compounds were ranked and 46 were selected for further experimental validation based on the total binding energy of the ligand/fibril complex. The experimental assessment consisted of direct measures of fibril disassembly (Fig. 2d, e and Supplementary Table 3) and assessment of the seeding capability of the products of disassembled AD tau fibrils in a biosensor cell assay[25]. Disaggregating molecules that produce active seeds are undesirable for therapy and are excluded from further study. HEK293T cells expressing CFP- or YFP-fused tau are transduced with exogenous AD tau fibrils. Externally applied AD-tau fibrils then induce the aggregation of the endogenous fluorescent tau, resulting in the formation of FRET-positive intracellular aggregates, visible as bright puncta by fluorescence microscopy (Fig. 2f). Automated image analysis of visible puncta provides an objective quantification of seeded aggregation. When AD-tau fibrils are pre-treated with compounds capable of disaggregation, the fibrils no longer seed the aggregation of the intracellular tau, and the cells remain diffusely fluorescent, with no visible puncta. Although biosensors may be limited in reporting structural characteristics of fibril progeny[26], they offer exquisitely sensitive measures of tau seeding activity—the ability for fibrils to recruit monomeric protein into fibrillar form, which is thought to be the mechanism for tau spreading in AD.

We identified 11 compounds that inhibit the seeding efficiency of crude AD brain extracts by at least 50%; 8 compounds from the CNS-Set library and 3 from the FDA-approved library (Fig. 2e). The top hit, FDA-A4 (Temoporfin), was an effective inhibitor but was toxic to biosensor cells under the assayed conditions. The activity of FDA-R20 (Phenylbutazone) was modest, so instead we focused our subsequent efforts on FDA-A2 (Lomitapide) and the remaining 8 best CNS-Set compounds by measuring dose-dependent inhibition of seeding by brain-purified AD-tau fibrils. As shown in Supplementary Fig. 7, dose-dependent inhibition of seeding is observed for 7 of these 9 compounds.

From the biosensor experiments, 4 of the CNS-Set compounds (CNS-11, 17, 2, and 12) inhibited seeding efficiency with $IC_{50}$ values <5 μM. These four lead compounds represent a relatively diverse chemical space, with a limited similarity between each compound or EGCG (Fig. 3a), and all with favorable drug-like qualities (Supplementary Table 4). Electron microscopy of tau fibrils treated with each compound reveals a qualitative reduction in fibrils, particularly for CNS-11 and CNS-12 (Fig. 3b). We then scrutinized the four CNS-Set inhibitors by quantitative EM (qEM) imaging (Fig. 3c, see "Methods"). Inhibitor CNS-11 stands out as having disaggregation activity approaching that of EGCG. CNS-12 also exhibited a reduction in AD-tau fibrils, although with a lower efficacy. As additional confirmation, we quantified the abundance of insoluble tau after treating AD-tau fibrils with disaggregating compounds using Western blot analysis (Fig. 3d, e). The results mirror qEM data with reductions in insoluble tau by CNS-11 and CNS-12. CNS-17 also showed a reduction in insoluble tau species, although a corresponding reduction in fibril density was not seen by qEM. The differences may reflect added shear forces that are exerted by SDS-PAGE/Western analysis, which disrupts fibrils that are chemically weakened, but not entirely disaggregated by bound disaggregants. By comparison, qEM is a gentler approach that is more reflective of the isolated effects of disaggregant binding. Lastly, using an MTT assay with Neuro 2a (N2a) cells, we assessed the neuronal cytotoxicity of each compound alone and after incubation with AD-tau fibrils (Fig. 3f). Apart from CNS-11, no significant cytotoxicity for EGCG

or the other lead compounds was detected. Importantly, no significant changes in cytotoxicity are observed for the samples after incubation with AD-tau fibrils, implying the disaggregated fibrils do not gain cytotoxicity.

The reduction of fibrils following EGCG treatment shown by negative-stain EM in Fig. 3b also rules out a possible alternative explanation for decreased detection of GT38 positive tau aggregation in the dot blot of Supplementary Fig. 1—namely interference of EGCG binding by antibody GT38. That is, the observed reduction of fibrils is produced by the disaggregating action of EGCG, not by the interference of EGCG with antibody GT38. Also supporting the action of EGCG as the cause of fibril disaggregation are the data of Fig. 2e, showing that EGCG treatment reduces seeding in biosensor cells.

## Molecular dynamics and energetics of compound–fibril complexes

To further understand how bound EGCG and lead compounds from the in silico screen may be destabilizing the tau PHF structure, we performed molecular dynamics (MD) simulations of each compound/fibril complex. Without a compound bound, the 4.8 Å spacing between layers of the fibril remains stable over the 100 ns duration of the MD simulation (Supplementary Fig. 8a). Docking a single EGCG molecule in the Site 1 binding cleft leads to an increase in separation between tau molecules to ~9 Å, which persists over the duration of MD simulation (Supplementary Fig. 8b). Inter-layer spacing was chiefly perturbed by EGCG at the segment spanning residues Lys340-Lys343 of AD-tau, particularly at Glu342 and Lys343. EGCG appears to rapidly form a hydrogen bond with the backbone amide of Ser341, disrupting the hydrogen-bond network between the fibril layers (Supplementary Fig. 8g, h). CNS-11 also destabilizes inter-layer spacing by MD, although the major perturbations are centered on Lys340 with additional perturbations seen at Ser341 and Glu342. None of the other CNS-set compounds perturbed the fibril in our simulation, consistent with our experimental results of CNS-11 and EGCG being the most effective disaggregants. These MD experiments suggest that an early contribution to tau fibril disaggregation by effective disaggregants is their competition for hydrogen bonds that bind adjacent tau molecules into tau fibrils.

The role of EGCG in disaggregating AD-tau is further highlighted by calculations of solvation-free energies of the unliganded and 3-h EGCG structures. Our estimates are based on free energies calculated using coordinates of the control and 3-h structures (Fig. 4a, b). The results show that each tau molecule in the fibril is less stable after 3-h EGCG incubation (−28.1 kcal/mol/chain) compared to the no-EGCG structure (−34.9 kcal/mol/chain). To identify which tau residues are most destabilized by EGCG binding, we subtracted the free energies for each atom of the control structure (more stable) from the 3-h structure with EGCG present (less stable). The resulting difference energy map reveals 2.5 kcal/mol decreased stability on Lys340, located within the EGCG binding site (Fig. 4c). This decrease is likely due to the burial of the positively charged Lys340 by EGCG. Subtle destabilization is evident in other residues lining the EGCG binding cleft (Fig. 4c). Together, these calculations demonstrate that the PHF structure captured after 3 h of EGCG incubation is overall less stable, with specific residues in the binding cleft contributing most strongly to the destabilization. Further detailed energetic analysis can be found in Supplementary Fig. 9.

## Discussion

Our cryogenically trapped structures of AD brain-extracted tau fibrils raise two questions related to the development of drugs for AD. The first is whether the tried-and-true structure-based methods of small-molecule drug discovery that have accelerated treatments for cancer and metabolic diseases can be effectively applied to Alzheimer's

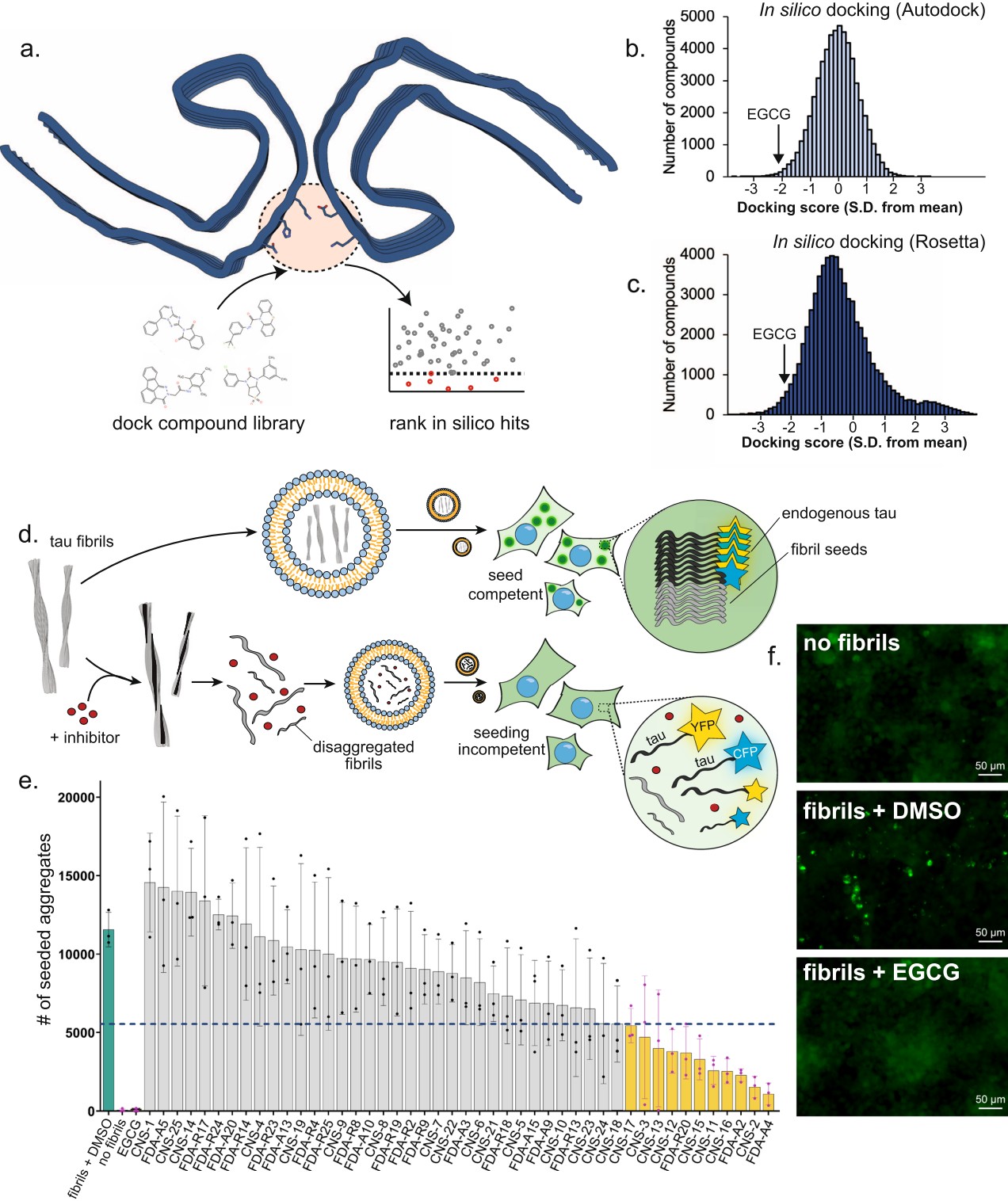

disease. The petty pace of drug development for Alzheimer's disease is rooted in part in the stable, traditionally intractable targets, amyloid fibrils. Amyloid fibrils lack the obvious concave active sites of enzymes and mobile helix interaction sites of GPCRs that offer binding cavities for drugs. Also frustrating progress has been the paucity of high-resolution amyloid structures so helpful for drug discovery.

Whereas our structure of 3-h EGCG-treated AD tau fibrils successfully identifies the useful EGCG pharmacophore, its 3.8 Å resolution falls short of atomic resolution, for several reasons. Chief among these is that we capture the EGCG pharmacophore during the kinetic course of its destruction. We may be averaging tau-EGCG particles that are at different stages of disassembly, leading to blurring in the final reconstruction. The consequent disorder hinders the cryoEM image alignment process. In fact, we were able to determine the less disturbed control and 1-h samples to 3.4 and 3.3 Å resolutions. Additionally, the occupancy of EGCG at Site 1 is only 66%, indicating that some fibril particles lack EGCG or are at less than full occupancy. This finding is similar to a recent study of PET ligands bound to AD tau having occupancies ranging from

**Fig. 2 | In silico and in vitro screening of tau disaggregants using EGCG pharmacophore. a** To identify novel compounds capable of fibril disaggregation, we performed an in silico screen using the EGCG binding site to the tau PHF (red circle). Two libraries of compounds were docked to the site using two computational methods (AutoDock and Rosetta), and hits were ranked and selected for experimental characterization. **b**, **c** Distribution of in silico docking scores of compound libraries using AutoDock (**b**) and Rosetta (**c**). For both methods, more negative scores indicate stronger compound binding. EGCG was a control for each method. As shown, both methods identify EGCG as a strong binder to the site on the tau PHF. **d** Top hits from the computational screen were selected for experimental characterization. Compounds were initially screened using an in vitro biosensor cell assay. Brain-derived tau fibrils were incubated with and without an inhibitor compound. Fibrils were then dissolved in liposomes and transduced into HEK293T cells overexpressing fluorescently labeled tau. When fibrils are transfected into the cells, the exogenous seeds initiate the aggregation of the endogenous tau, resulting in the formation of intracellular fluorescent puncta. If fibrils are effectively disaggregated by an inhibitor compound, the exogenous fibril seeds will be dissolved, and the intracellular tau will remain soluble, with no puncta formed. **e** Quantification of hit compounds in tau biosensor cell assay. For fibrils treated with DMSO vehicle control (turquoise bar), many fluorescent aggregates are seen. Without the addition of fibrils, ("no seed"), no intracellular aggregation occurs. Incubation of fibrils with EGCG also prevents the formation of any seeds. The dashed line indicates a 50% reduction in the number of aggregates. Yellow bars indicate any compound that produces a >50% reduction in aggregate formation. Error bars represent ±SD, all experiments were performed with $n = 3$ experimental replicates. **f** Fluorescent microscopy images of biosensor cells without fibril seeds added (top), with seeds and DMSO control (middle), and with seeds and EGCG (bottom). Numerous bright intracellular puncta are seen in the DMSO control, which are eliminated with the addition of EGCG.

~40–70%[27]. Yet despite the limited resolution, the wider implication of the EGCG-AD-tau structure is that structure-based drug discovery is possible for amyloid diseases, as it has been for other medical conditions.

In order for structure-based drug discovery efforts for AD to be as successful as efforts in other diseases, disaggregants identified as drug leads must be free of toxicity at effective concentrations and must produce products of disaggregation that are not toxic and which do not seed monomeric tau into amyloid fibrils. The natural cellular machinery including the chaperone[28] and ubiquitin ligase systems[29] evolved to disassemble aggregated proteins. Aids are needed since these systems are quickly overrun as aggregate burden increases and may also exacerbate tau aggregation in the process[30,31]. We use tau biosensor seeding as a screening assay to identify compounds that produce non-seeding competent products. We note this assay is not definitive because presently available tau biosensor cells, although sensitive to seeding by a range of tau structures, produce tau aggregates that probably differ in polymorphic form from AD-tau. Absolute assurance that the disaggregated products of AD-tau are not seeding competent in the brain awaits further experimentation.

The other major question raised by our structures is how a small molecule can dismantle extraordinarily stable pathogenic amyloid fibrils. Our structure of EGCG bound to AD-tau suggests two possible mechanisms, which may operate in concert. The first we term the charge-pairing mechanism. It is based on favorable EGCG-tau interactions that destabilize stacked tau molecules. The second we term the EGCG curvature mechanism based on favorable EGCG–EGCG interactions that pry apart the fibril.

Charge-pairing is a well-known stabilizing feature of amyloid fibrils, and its disruption can lead to charge repulsion by like charges stacked in adjacent fibril layers[32]. Examination of the 3-h structure shows that EGCG H-bonds to side chains of tau in the two clefts (Fig. 1d and Supplementary Fig. 10). This can block Lys340 from charge pairing with neighboring Glu338 and Glu342. Once the positive charge of Lys340 no longer compensates the negative charges on the Glu residues, the Glu residues will repel the Glu residues in adjacent layers of tau molecules, weakening the fibril. Our energy difference maps support that EGCG destabilizes the fibril as they show a large increase in the energy of Lys340 and the surrounding residues of the EGCG pharmacophore. Tau destabilization may also occur at EGCG Sites 2 and 3, as both sites are adjacent to complementary lysine/aspartic acid charge pairs that may be similarly disrupted by EGCG binding. The disruption of charge pairing may be an early stage of fibril disassembly as it creates repulsion between layers of the fibril, potentially setting the stage for tau molecules to begin separating. As adjacent tau molecules begin to separate, our MD simulations demonstrate that destabilization between tau molecules is energetically compensated by hydrogen bonds between EGCG and the backbone amide of Ser341 (Supplementary Fig. 8). One of the most effective EGCG analogs that we discovered in our screen, CNS-11, is calculated to interact with AD-tau by many of the same hydrogen-bonding interactions as EGCG, including potential interactions with Lys340 and Glu342 (Fig. 3g).

In our second proposed mechanism, interactions between stacked EGCG molecules destabilize fibril integrity. Our structure of EGCG bound to AD-tau reveals EGCG molecules stacked 4.8 Å apart, which permits each EGCG molecule to hydrogen-bond with individual stacked molecules of tau (Fig. 4d). However, this 4.8 Å spacing incurs unfavorable voids between EGCG rings A, C, and D of neighboring molecules in the stacks (gaps between space-filling atoms in Fig. 4d). The stack of EGCG molecules can be stabilized by compressing the distance between the solvent-facing aromatic rings to about 3.5 Å, which achieves van der Waals contact. This compression incurs curvature of the stack and widens the spacing on the fibril-facing surface (Fig. 4e). This curving effect is also observed in an MD simulation of a 5-layer stack of EGCG molecules (Supplementary Fig. 11), and is reminiscent of the tilted interaction between EGCG molecules in the form IV EGCG crystal structure[22]. Asn327 and His329 can maintain favorable interactions with the curved stack of EGCG molecules only if the tau molecules separate. This separation could allow water to solvate the separated tau molecules and enable other EGCG molecules to invade as seen in our MD simulation. By this mechanism, the binding energy between stacked EGCG molecules is converted to a conformational change that pries apart stacked tau molecules (Fig. 4f).

The molecular events described by these two disaggregation mechanisms can be unified in a reaction coordinate diagram (Fig. 4g). First, EGCG binds to the fibril, incurring entropy loss due to its ordering along the fibril surface (Fig. 4g, step B). Next, a quasi-stable intermediate state evolves, represented by the 3-h structure (Fig. 4g, step C). The intermediate is stabilized by stacking interactions between EGCG molecules and hydrogen bonds between EGCG and the binding cleft. Importantly, the energy well is not inescapably deep, owing to destabilization incurred by EGCG disruption of charge-pairing and the trapping of small voids created between stacked EGCG molecules. Next, tau molecules are pried apart by charge repulsion incurred by EGCG, and curvature induced by the elimination of voids between stacked EGCG molecules (Fig. 4g, step D). Lastly, solvent and additional EGCG molecules invade the gaps between tau molecules, completing the disaggregation of tau (Fig. 4g, step E). Thus, our structure offers insight into how a small molecule can dismantle an extraordinarily stable pathogenic amyloid fibril.

We note that the kinetics of disaggregation are relatively slow, particularly for a process that involves high-affinity binding of EGCG and breaking of hydrogen bonds. We attribute slow kinetics to our proposed mechanism of disaggregation (Fig. 4d), in which cumulative binding of EGCG causes a concerted conformation change in stacked

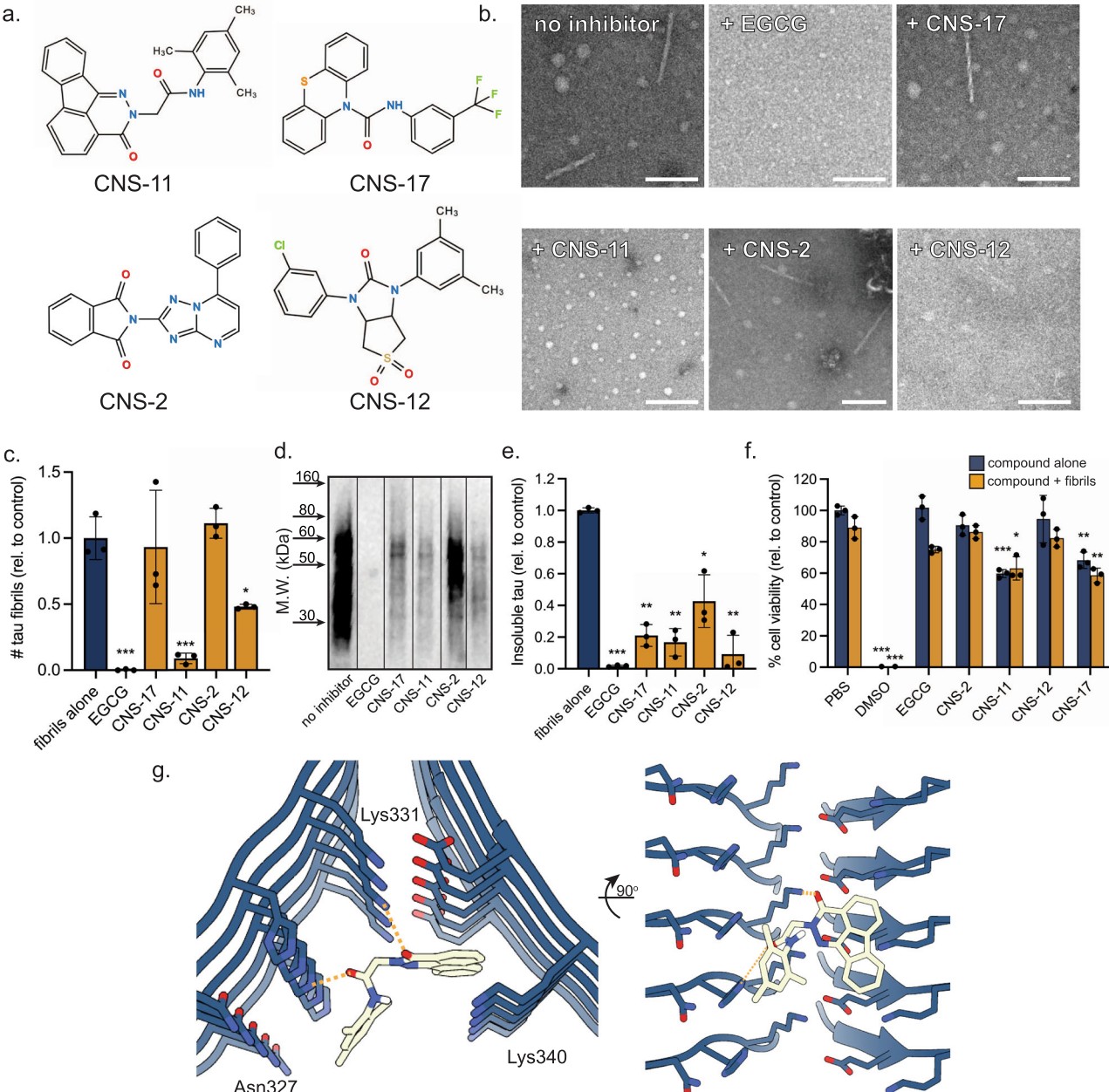

**Fig. 3 | Characterization of tau disaggregation by lead compounds. a** Top hits from the in silico and biosensor cell screens were selected for further experimental characterization. Four compounds were selected, CNS-11, CNS-17, CNS-2, and CNS-12. **b** Electron micrographs of brain-derived tau fibrils after incubation with each compound, with EGCG as control. Few fibrils are observed with EGCG treatment, as well as with CNS-11 and CNS-12. Scale bars represent 250 nm. **c** Quantitation of fibril number present on EM images with and without compound treatment. $N = 33$ images were taken from random points on the EM grid, and fibrils were counted. Error bars represent the standard deviation of triplicate technical measurements. A large reduction in visible fibrils is seen for CNS-11. **d** Brain-derived tau fibrils were treated with compound and the insoluble fraction was analyzed by Western blot,

staining for total tau. **e** Quantitation of insoluble tau abundance in the Western blot. Similarly, both EGCG and the lead compounds substantially reduce the amount of insoluble tau in the fraction. $N = 3$ experimental replicates were performed for each treatment condition. **f** MTT cytotoxicity assay in Neuro2a cell model. Brain-derived tau PHFs with and without vehicle control (PBS) show no toxicity (blue bars). Compounds alone show varied toxicities (dark orange bars). Compounds incubated with tau fibrils (light orange bars) do not show additional toxicity. All error bars represent ±SD. $N = 3$ experimental replicates were performed for each **g** Model of CNS-11 docked to the EGCG binding site on the tau PHF. CNS-11 is within the hydrogen bonding distance of both H329 and K340. For **c, e, f** $*p < 0.05$, $**p < 0.01$, $***p < 0.001$ using a one-way ANOVA.

columns of EGCG, forcing tau layers apart. We suggest this process of conformation equilibrium where forces required to remodel the fibril are counterbalanced by the physiochemical properties of EGCG is a kinetically slow process.

Our proposed mechanism of EGCG-mediated AD tau fibril disaggregation is not necessarily specific to tau fibrils. EGCG is known to disassemble fibrils composed of many different amyloid proteins[7,11]; its effects are not sequence specific, but are instead specific to the

amyloid scaffold itself. The hydroxyl-rich EGCG molecule likely seeks out a charge-rich region of any given amyloid fibril, burying charge and stacking planarly alongside it to produce conformational stress that eventually disassembles the fibril. The promiscuity of EGCG highlights the need in the future to not only identify analogs with better drug-like properties, as we have focused on here, but also to make analogs that are amyloid-specific in order to avoid disaggregating amyloids that form as part of normal biological processes[33].

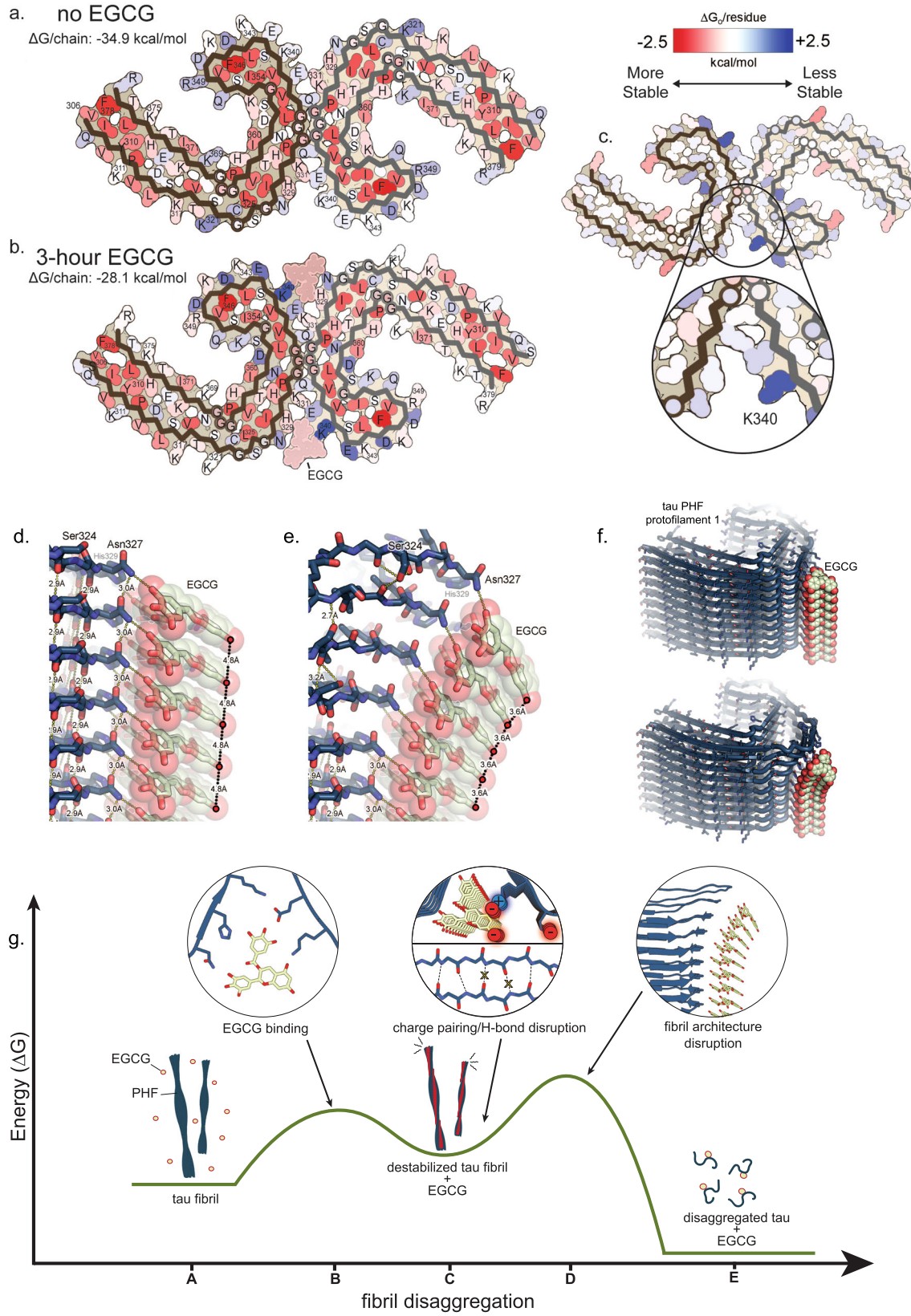

Here we have determined the intermediate structure of AD tau fibrils on the pathway to EGCG-driven disaggregation. Considering the EGCG binding site a pharmacophore for disaggregating compounds, we identified several tau-disaggregating molecules with physiochemical drug-like properties superior to those of EGCG. Further screening or optimization of these compounds may result in a new generation of AD drug leads capable of entering neurons and effectively disaggregating tau fibrils into inert products. Our approach may be applicable to amyloid proteins involved in other amyloid-based degenerative conditions such as Parkinson's disease and systemic amyloidosis.

**Fig. 4 | Structure-informed mechanism for EGCG-driven disaggregation of AD-tau PHF.** Solvation energy calculations of tau PHF structures without EGCG (**a**) and with EGCG after 3-h incubation (**b**). Red residues are more stable; blue residues are less stable. The most stable residues seen across both structures are hydrophobic and buried within the fibril core, and less stable residues are typically on the solvent-exposed surface. At 3-h incubation, the structure is less stable (−28.1 kcal/mol/chain) than without EGCG (−34.9 kcal/mol/chain). **c** To understand the localized effects in fibril stability, energy difference maps were calculated. Subtraction of the no-EGCG model from the 3-h EGCG model shows a large increase in free energy of Lys340 at the EGCG binding site, indicating the presence of EGCG significantly destabilizes Lys340. **d** The 3-h PHF-EGCG structure reveals EGCG molecules stacked 4.8 Å apart, permitting each EGCG molecule to H-bond with individual stacked molecules of tau (dashed yellow lines connecting EGCG to tau side chains Asn327 and His329). The 4.8 Å spacing between tau molecules is characteristic of the intermolecular β-sheet hydrogen bonding distance (dashed yellow lines connecting tau molecules). However, this 4.8 Å spacing incurs unfavorable voids between EGCG rings A, C, and D (as indicated by gaps between space-filling atoms). **e** The voids between stacked aromatic groups can be filled by compressing the distance between these A, C, and D aromatic rings that face the solvent. In so

doing, the EGCG stack curves, widening the spacing on the fibril-facing surface. Asn327 and His329 can maintain favorable hydrogen bonding with the curved stack of EGCG molecules only if the tau molecules separate wider than 4.8 Å. This separation would allow water to solvate the separated tau molecules. The curvature of the EGCG stack fills the unfavorable voids between EGCG aromatic rings and further widens the separation between tau molecules. By this mechanism, the binding energy between stacked EGCG molecules is converted to a conformational change that pries apart stacked tau molecules. **f** Alternate view showing a tau PHF protofilament being disrupted by the curvature of stacked EGCG molecules. **g** Reaction coordinate diagram describing the possible mechanism of tau disaggregation by EGCG. Tau PHFs in solution with EGCG (coordinate A) are bound by repeating stacks of EGCG molecules (coordinate B). Once EGCG is bound, local charge-mediated effects begin to destabilize the fibril (coordinate C). These effects include unfavorable burying of charged residues (e.g. Lys340) and disruption of pairing between charged side chains. These repulsive forces, in addition to possible backbone H-bonding between tau and EGCG, weaken the β-sheet H-bond network of the fibril. Lastly, conformational changes induced by EGCG π-π stacking (described in **d**–**f**) may further disrupt the fibril architecture (coordinate D), leading to the disaggregated EGCG-bound tau end product (coordinate E).

## Methods

### Preparation of crude and purified brain-derived tau seeds

Human autopsy samples were obtained by the Mayo Clinic Brain Bank and UCLA Pathology Department according to HHS regulations from patients consenting to autopsy. Samples were provided to the researchers of this study as anonymized tissues. For purification of paired helical filaments (PHFs) and straight filaments (SFs) from AD brain tissue, extractions were performed according to the previously published protocol without any modifications[21]. Prior to cryoEM grid preparation, AD brain-purified tau fibrils were pre-incubated at 37 °C with 0.5 mM EGCG that was dissolved in PBS in a buffer comprised of 20 mM Tris–HCl pH 7.4, 100 mM NaCl. Control fibrils from the same brain donor were treated identically except for the addition of EGCG. For biosensor seeding assays with crude AD brain extracts, fresh-frozen tissue from neuropathologically confirmed AD cases was thawed and cut into a 0.2–0.3 g sections. Tissue was manually homogenized in a 15 ml disposable tube in 1 ml of 50 mM Tris, pH 7.4 with 150 mM NaCl, and then aliquoted to PCR tubes and sonicated in a cuphorn bath for 120 min under 30% power at 4 °C in a recirculating ice water bath, according to reference[17].

### CryoEM data collection and helical reconstruction

AD brain purified tau fibrils were applied to glow-discharged Quantifoil 1.2/1.3 electron microscope grids (2.6 µl for 1 min), and subsequently plunge-frozen in liquid ethane on a Vitrobot Mark IV (FEI). Data were collected on a Titan Krios (FEI) microscope (operated with 300 kV acceleration voltage and a slit width of 20 eV); the 3-h EGCG pre-incubation dataset was collected with a Gatan Quantum LS/K2 Summit direct electron detection camera. Control and 1-h EGCG pre-incubation datasets were collected using the Gatan K3 BioQuantum direct electron detection camera. Counting mode movies were obtained with a nominal physical pixel size of 1.07 Å per pixel with a dose per frame 0.63 e⁻/Å² (K2 data set) with a total of 30 frames at a frame rate of 5 Hz, resulting in a final dose 19 e⁻/Å² per image. For K3 datasets, the physical pixel size was 0.549 Å per pixel. The dose per frame for the control data set was 1.66 e⁻/Å² with a total of 48 frames at a frame rate of 20 Hz resulting in a final dose 80 e⁻/Å² per image, and for the 3-h EGCG pre-incubation dataset the dose per frame was 1.2 e⁻/Å² at a frame rate of 20 Hz resulting in a final dose 56 e⁻/Å² per image. Automated data collection was driven by the Leginon 3.3 automation software package[34].

All datasets were pre-processed using Unblur 1.00[35] and CTFFIND 4.1.8[36]. The 3-h EGCG data set was corrected for magnification anisotropy using Unblur using anisotropy parameters generated from mag_distortion_estimate 0.0.0[37] performed on crystalline ice images. Helical tubes were picked manually using EMAN2.2 e2helixboxer.py[38].

For the control data set, particles were extracted with a 320-pixel box size and 10% interbox distance. 2D and 3D classification as well as gold-standard refinement were performed using Relion 3.1[39]. A featureless cylinder was used as an initial 3D model. For the 1- and 3-h data sets, the following data processing strategy was employed. Initial particles were extracted using a 686-pixel box size with a 10% inter-box distance and downscaled by 2. After 2D and 3D classification using a featureless cylinder as an initial model, particles were re-extracted using a 432-pixel box size for the 3-h data set and 320-pixel box size for the 1-h data set without downscaling and refined to high resolution using further rounds of 3D classification and gold-standard refinement. The no EGCG and 3-h reconstructions were sharpened adhoc using RELION postprocess while the 1-h structure was auto-sharpened using phenix.auto_sharpen (Phenix 1.2)[40].

### Occupancy analysis

To calculate the occupancy of EGCG density relative to the fibril, we applied three masks over the solvent, over residues 345–352 of the protein backbone, and EGCG Site 1 density. Using the voxel values within the three masks, the EGCG occupancy was calculated according to the following formula:

$$\text{Occupancy} = (\max(\text{EGCG}) - \text{avg(solvent)}) / (\max(345 - 352) - \text{avg(solvent)}).$$

### Atomic model building

The highest-resolution structure of an AD-brain derived tau PHF (PDB 6HRE) was used as a starting model for atomic model building. 6HRE coordinates were first docked in the 3.8 Å density map of the 3-h reconstruction as a rigid body, and subsequent manual refinement was performed in COOT 0.9.8.2[41]. EGCG was modeled into the density by first rigid body docking molecule KDH 911 of PDB 4AWM and subsequent real-space refinement. Five fibril layers were added according to the symmetry of the helical reconstruction to maintain local contacts between chains in the fibril during structure refinement. We performed automated structure refinement using phenix.real_space_refine (Phenix 1.2)[42]. A similar process was performed for the no EGCG control structure, omitting EGCG. The no EGCG fibril structure was rigid body fit into the 1-h density to achieve the 1-h structure.

Additional poses of EGCG were considered by manually placing six plausible alternate conformations of EGCG in binding Site 1. After manual and automatic refinement using phenix.real_space_refine, poses were evaluated using: model validation statistics, buried surface area, shape complementarity, and free energy[43].

CryoEM data and model refinement statistics are presented in Supplementary Table 1. Half-map/half-map FSC and model/map FSC curves are shown in Supplementary Fig. 12.

### Negative stain grid preparation

Negatively stained EM Grids were prepared by depositing 6 μl of sample on formvar/carbon-coated copper grids (400 mesh) for 3 min. The sample was rapidly wicked using filter paper without drying the grid and stained with 1% uranyl acetate for 2 min. For quantitative EM image (qEM), Negative-stain EM grids of each sample were screened at a magnification of ×11,500, collecting images in 5-micron increments. Fibrils were counted from collections of 33 micrographs (in triplicate) for each experimental condition.

### Solvation-free energy calculations

Solvation free energy calculations were performed as recently described[43]. Briefly, the solvent accessible surface area (SASA) for each atom on a central strand within an amyloid fibril was determined (folded state). Next, the SASA for each atom of the isolated extended strand was determined (reference state), and the difference between both states was calculated ($SASA_{Ref} - SASA_{Fold}$). This value was then multiplied by the Atomic Solvation Parameter (ASP) specific to each atom, as determined previously by the authors in ref. 44. An entropic term is also included to take into consideration the degrees of freedom lost in going from a disordered to an ordered state[45]. The energies of all atoms were then summed to generate the solvation energy for each structure. Difference energy maps were generated by subtracting solvation-free energies pairwise for each atom in the two structures being compared.

### In silico docking

Docking calculations were performed using two separate methodologies, AutoDock Vina[23] and RosettaLigand[24]. The two compound libraries used for docking were the FDA-approved and CNS-Set from Chembridge. Two-dimensional compound coordinates were downloaded and converted to 3D using Open Babel[46]. For AutoDock Vina, version 1.1.2 was used, and all parameters were kept at default values. A region at the EGCG Site 1 of the tau PHF was defined as the docking site using a 20 Å × 16 Å × 12 Å box. Compounds were ranked by binding energy. For RosettaLigand, once three-dimensional ligand structures were generated using Open Babel, we generated a ligand perturbation ensemble using our previously described method[47]. For each rotatable bond of the ligand, a torsion angle deviation of ±5° was applied, generating 100 conformations for each ligand. Ligand docking was performed using the HighResDocker mover in the Rosetta Scripts modality (Rosetta version 3.10), using the Ref2015 energy function. A 7 Å box centered at the EGCG Site 1 binding site on the tau PHF was used as the docking site. Cycles of side chain repacking were coupled with small ligand perturbations every third cycle. Ligand poses were ranked based on the lowest binding energy, and interface energies were calculated using the InterfaceScoreCalculator mover.

### Molecular dynamics simulations

MD simulations were performed using GROMACS version 2018[48] and the CHARMM27[49] all-atom forcefield. The 3-h tau fibril EGCG complex structure with 10 protein monomers was solvated in a cubic water box using periodic boundary conditions with counter ions added. Systems were energy minimized then temperature and pressure equilibrated for 100 ps. Production runs were carried out for 100 ns. Calculations of non-bonded interactions were gpu accelerated. Calculations were performed with and without bound small molecules. Small molecules were placed into the EGCG Site 1 binding site, with their initial conformations determined by the output from the AutoDock Vina binding (see above). Ligand

topologies were calculated using the CHARMM General Force Field server (CgenFF), and hydrogens were added to ligands using Avagadro.

### Inhibitor screening in tau biosensor cells

HEK293T cell lines stably expressing tau-K18 P301S-eYFP were obtained from Marc Diamond[25] and used without further characterization or authentication. Cells were maintained in DMEM (Life Technologies, cat. 11965092) supplemented with 10% (vol/vol) FBS (Life Technologies, cat. A3160401), 1% penicillin/streptomycin (Life Technologies, cat. 15140122), and 1% Glutamax (Life Technologies, cat. 35050061) at 37 °C, 5% $CO_2$ in a humidified incubator. Fibrils and patient-derived crude brain extracts were incubated for 16–18 h at 4 °C with indicated inhibitor to yield a final inhibitor concentration of 10 μM (on the biosensor cells), except for IC50 determinations, which instead used adjustments to achieve the final indicated inhibitor concentration. Inhibitors were dissolved in DMSO. For seeding, inhibitor-treated seeds were sonicated in a cuphorn water bath for 3 min, and then mixed with 1 volume of Lipofectamine 3000 (Life Technologies, cat. 11668027) prepared by diluting 1 μl of Lipofectamine in 19 μl of OptiMEM. After twenty minutes, 10 μl of fibrils were added to 90 μl of tau biosensor cells. The number of seeded aggregates was determined by imaging the entire well of a 96-well plate in triplicate using a Celigo Image Cytometer (Nexcelom) in the YFP channel. The number of aggregates in a given image was determined using an ImageJ 2.3.0[50] script, which subtracts the background fluorescence from unseeded cells, and then counts the number of aggregates as peaks with fluorescence above the background using the built-in Particle Analyzer. The number of aggregates was normalized to the confluence of each well, and dose-response plots were generated by calculating the average and standard deviations from triplicate measurements.

### MTT cytotoxicity assay

In all, 90 μL of Neuro2a cells were plated on clear 96 well plates (Falcon 353072) at 6000 cells per well and given 24 h to adhere to the plate. Subsequent treatment consisted of PHF samples that were co-incubated with or without 100 μM of small molecules for 48 h at 37 °C, with 10 μl applied per well for a final small molecule concentration of 10 μM. After another 24-h incubation at 37 °C, 20 μl of Thiazolyl Blue Tetrazolium Bromide MTT dye (Sigma; 5 mg/ml stock in DPBS) was added to each well and further incubated for 3.5 h at 37 °C. The assay was arrested by replacement of all media with 100% DMSO and transferred to a SpectraMax MF reader. Background 700 nm readings were subtracted from the absorbance readings at 570 nm. Cells treated with 100% DMSO were designated as 0% viable and those treated only with vehicle were designated as 100% viable. Other well readings were normalized to these values.

### Western blot analysis

AD brain-derived tau fibrils were incubated with compounds for 72 h at 37 °C in PBS. Compounds were centrifuged at $21,130 \times g$ for 60 min to separate soluble and insoluble fractions. Samples were loaded onto gels (NuPAGE 12% Bis-Tris pre-cast) and ran at 200 V for 40 min. Proteins were transferred from gel to nitrocellulose membrane using an iBLOT2 system. The membrane was blocked for 1 h in TBST with 5% milk and washed with TBST three times. The membrane was then incubated with primary antibody (anti-tau A0024 (Dako), 1:4000 dilution in 2% milk/TBST solution) for 1 h, washed three times with TBST, incubated with horseradish peroxidase-conjugated secondary antibody (goat anti-mouse IgG H and L (HRP); 1:5000 Dilution in 2% milk/TBST), and washed three times in TBST. Signal was detected with Pierce ECL Plus Western Blotting Substrate (Cat # 32132), and the blot was imaged with a Pharos FX Plus Molecular Imager.

## Dot blot analysis

Brain-derived tau PHFs were incubated with 80 μM EGCG at 37 °C in PBS for various lengths of time. Samples were spotted onto nitrocellulose membrane, 20 μl were spotted for each condition, spotting 2 μl at a time and allowing to dry in between. The membrane was blocked for 1 h in TBST with 5% milk and washed with TBST three times. The membrane was then incubated with primary antibody (either anti-tau GT38 (obtained from Virginia Lee lab at UPenn) or anti-phospho-tau AT8 (cat # MN1020, lot # UL2906281Z), 1:4000 dilution in 2% milk/TBST solution) for 1 h, washed three times with TBST, incubated with horseradish peroxidase-conjugated secondary antibody (goat anti-mouse IgG H and L (HRP); 1:5000 Dilution in 2% milk/TBST), and washed three times in TBST. Signal was detected with Pierce ECL Plus Western Blotting Substrate (cat # 32132), and the blot was imaged with a Pharos FX Plus Molecular Imager.

## Reporting summary

Further information on research design is available in the Nature Research Reporting Summary linked to this article.

## Data availability

Structure coordinates and map files are deposited into the Worldwide Protein Data Bank (wwPDB) and the Electron Microscopy Data Bank (EMDB) with the following accession codes: PDB ID 7UPE/EMD-26663 (no EGCG), PDB ID 7UPF/EMD-26664 (1 h), and PDB ID 7UPG/EMD-26665 (3 h). Source data are provided with this paper.

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

## Acknowledgements

We thank the donors and their families; without whom this work would not have been possible. We acknowledge NIH 1R01AG070895 (to D.S.E.) and NIH R01 AG048120 (to D.S.E.) from the National Institute on Aging, and Howard Hughes Medical Institute (to D.S.E.); 1F32 NS095661 from the National Institute of Neurological Disorders and Stroke (to P.M.S.), A2016588F from the BrightFocus Foundation (to P.M.S.), GM08042 from the UCLA-Caltech Medical Scientist Training Program (to K.A.M.), the UCLA Chemistry-Biology Interface training grant (USPHS National Research Service Award 5T32GM008496) (to K.A.M.), and HHMI for support. We thank Jeffrey Zhang for discussion and Garrett Gibbons and Virginia Lee for generously gifting the GT38 antibody. We acknowledge the use of instruments at the Electron Imaging Center for NanoMachines supported by NIH (1S10RR23057, 1S10OD018111, and 1U24GM116792), NSF (DBI-1338135), and CNSI at UCLA.

## Author contributions

The project was conceived and designed by P.M.S., K.A.M., D.R.B., and D.S.E. Acquisition and histologic/diagnostic analysis of patient brain tissue were performed by M.A.D., D.W.D., C.K.W., and H.V.V. P.M.S., K.A.M., X.C., and C.J.H. extracted tau fibrils from patient brains. CryoEM grids were prepared, and data were collected by P.M.S. with assistance from P.G. P.G., D.R.B., and P.M.S. processed cryoEM data and built atomic models with assistance from M.R.S. K.A.M. carried out in silico compound screening. In vitro compound screening assays were performed by K.A.M., P.M.S., H.P., C.J.H., and R.A. X.C. and C.J.H. performed dot blot analysis. D.R.B. and M.R.S. calculated solvation energies. K.A.M. performed molecular dynamics simulations. The manuscript was prepared by K.A.M., D.R.B., P.M.S., M.R.S., and D.S.E. with contributions from all other authors.

## Competing interests

D.S.E. is SAB chair and equity holder of ADRx, Inc. All other authors declare no conflicts.
