## [Peer Review File · Nature Communications]

Structure-based discovery of small molecules that disaggregate Alzheimer's disease tissue derived tau fibrils in vitroEditorial Note: This manuscript has been previously reviewed at another journal that is not operating a transparent peer review scheme. This document only contains reviewer comments and rebuttal letters for versions considered at *Nature Communications*.

REVIEWERS' COMMENTS

Reviewer #1 (Remarks to the Author):

The manuscript by Seidler, Murray and Boyer et al. has undergone significant revision and is greatly improved as a result. Through the revision process the focus of this manuscript has shifted from the mode of action of EGCG on tau fibers to a broader demonstration of the application of these structure-derived insights to the development of small molecules amyloid “disaggregation agents”. The new structures of untreated AD-tau fibers and those incubated with EGCG for 1 hour provide additional evidence for the proposed binding site of EGCG, while the in-silico screening for small molecules with disaggregation activity demonstrates the usefulness of these insights. Previous concerns have been addressed by these additional experiments as well as the shift in focus of the study.

Only a few minor questions remain and once addressed, this reviewer would support the publication of the revised manuscript in nature communications:

- 1/ Can the possible interference of EGCG with antibody (GT38) binding be excluded as an explanation for the decreased detection of GT38 positive tau aggregates in the dot-blot experiments?
- 2/ How do the authors explain the relatively slow kinetics of disaggregation? The fact that a ligand bound structure of the AD-tau fibers could be obtained implies EGCG binds tau fibrils with relatively high affinity and the suggested mechanism based on hydrogen-bond network disruption is not obviously a slow process.
- 3/ The structure of EGCG bound AD-tau fibers presented here is an in vitro reconstitution of this complex where the interaction, and the disaggregation mechanism, is primarily driven by disrupting the hydrogen-bond network between fibril layers. Would components of the cellular environment (salts, metal ions) affect the potential potency of EGCG or CNS11 in vivo?
- 4/ A reference in the introduction is missing ([ref])

Reviewer #2 (Remarks to the Author):

This is mostly OK now. The manuscript has improved significantly and I support publication.

Remaining (minor) points that would make the work stronger (but I don't need to see this again):

- 1) calculate difference maps between 0hr and 3hr maps and report sigma-levels of the difference densities, just as one would do in Xray crystallography (this was already suggested in my original review, but ignored). It would also be useful to see XY-cross sections of the density (possibly summed over one beta-rung) to better assess the reconstructed densities. Ext Data Figure 2 suggests that the 3hr density is rather poor...
- 2) The discussion mentions the pixel size and the electron scattering factors of the compound as possible reasons for lower resolution. This is highly unlikely and I suggest removal of this sentence. It is most likely that the resolution is limited by less order in the filaments.

Point-by-point response to reviewers' comments:

Reviewer 1:

1/ Can the possible interference of EGCG with antibody (GT38) binding be excluded as an explanation for the decreased detection of GT38 positive tau aggregates in the dot-blot experiments?

Response: Good question. This alternative explanation of interference of EGCG with antibody GT38 for the decreased detection of GT38 positive tau aggregates is ruled out by the observed reduction of fibrils following EGCG treatment by negative-staining EM, as shown in Fig. 3b. Also supporting the action of EGCG as the cause of fibril disaggregation are the data of Fig. 2e, showing that EGCG treatment reduces seeding in biosensor cells.

We have added a short paragraph explaining this in the revised text at the end of the section titled Screening for tau disaggregants using the EGCG pharmacophore.

2/ How do the authors explain the relatively slow kinetics of disaggregation? The fact that a ligand bound structure of the AD-tau fibers could be obtained implies EGCG binds tau fibrils with relatively high affinity and the suggested mechanism based on hydrogen-bond network disruption is not obviously a slow process.

Response: We explain the relatively slow kinetics of disaggregation by our postulated mechanism discussed in the sixth paragraph of the Discussion, namely that the cumulative binding of EGCG causes a concerted conformational change in stacked columns of EGCG, forcing tau layers apart. This seems likely to be a slow process, despite the high affinity of EGCG for tau fibrils.

In the revised ms, we have inserted a short paragraph on the slow kinetics, the third to last paragraph of the Discussion.

3/ The structure of EGCG bound AD-tau fibers presented here is an in vitro reconstitution of this complex where the interaction, and the disaggregation mechanism, is primarily driven by disrupting the hydrogen-bond network between fibril layers. Would components of the cellular environment (salts, metal ions) affect the potential potency of EGCG or CNS11 in vivo?

Response: It is a near certainty that components of the cellular environment affect the potency of EGCG and possibly also that of CNS-11, which although it has more drug-like properties than EGCG, also displays a lower IC50 than EGCG. We are currently screening for drug-like compounds with better potency than CNS-11, but this involves new methods that go beyond those of the current ms.

4/ A reference in the introduction is missing ('[ref]')

Response: This omission has been repaired in the revised ms.

Reviewer 2:

1) calculate difference maps between 0hr and 3hr maps and report sigma-levels of the difference densities, just as one would do in Xray crystallography (this was already suggested in my original review, but ignored). It would also be useful to see XY-cross sections of the density (possibly summed over one beta-rung) to better assess the reconstructed densities. Ext Data Figure 2 suggests that the 3hr density is rather poor...

Response: The requested difference map has been added to Extended Data Fig. 3 as panel d, with a corresponding description in the figure legend. The difference map reinforces our conclusions about the structures, showing EGCG density in the difference map as 3 sigma above the noise. A sentence noting this addition has been added to the second paragraph of Results in the revised ms.

Of importance, the densities found flanking K311/K317/K321 that are present in all AD tau fibril structures, including our 0 hr. and 3 hr. structures (densities seen at bottom left and top right of the cross-sections) are eliminated in the difference map. This demonstrates that we captured known additional densities in our structures and that they are equally present in both our 0 and 3 hr. structures, serving as an internal control.

2) The discussion mentions the pixel size and the electron scattering factors of the compound as possible reasons for lower resolution. This is highly unlikely and I suggest removal of this sentence. It is most likely that the resolution is limited by less order in the filaments.

Response: The “unlikely” sentence has been removed, as suggested.